# The Effect of the “Colorful Eating Is Healthy Eating” Long-Term Nutrition Education Program for 3- to 6-Year-Olds on Eating Habits in the Family and Parental Nutrition Knowledge

**DOI:** 10.3390/ijerph19041981

**Published:** 2022-02-10

**Authors:** Malgorzata Kostecka

**Affiliations:** Faculty of Food Science and Biotechnology, University of Life Sciences in Lublin, Akademicka 15, 20950 Lublin, Poland; julka-portal@wp.pl or kostecka.malgorzatam@gmail.com; Tel.: +48-81-445-6846

**Keywords:** preschool children, multi-stage education program, nutrition knowledge, parents’, eating behaviors, nutrition quality

## Abstract

Background: Effective strategies for improving eating habits and dietary intake in preschoolers are essential for reducing the risk of chronic non-infectious diseases in later life. The aim of this study was to evaluate the effect of long-term nutrition education for 3- to 6-year-olds on parental nutrition knowledge. Methods: The study was conducted as part of the “Colorful Eating is Healthy Eating” nutrition education program that has been implemented in kindergartens in Lublin since 2016. A total of 11 kindergartens were involved in this stage of the program, and 733 parents consented to participate in the project. The study was divided into three stages. In the first stage all parents completed a questionnaire containing 54 items. In the next stage, 211 children from four randomly selected kindergartens participated in the “Colorful Eating Is Healthy Eating” educational program that lasted 7 months. In the third stage of the study, the parents of children who had completed the 7-month educational program and the parents of control group children once again completed the questionnaire. Results: A positive outcome of the educational program was that it contributed to a decrease in the consumption of sweetened hot beverages (*p* = 0.005) and an increase in water intake (*p* = 0.001). The nutrition education program was also successful in reducing the consumption of sweets. Children’s education improved the parents’ knowledge about dietary sources of fiber and the recommended fiber intake, and it contributed to the awareness that breakfast is the most important meal of the day. The program did not enhance the parents’ knowledge about snacking between meals or the role of sweetened beverages in dental caries, overweight and obesity. Conclusions: Long-term multi-stage nutrition education for children aged 3 to 6 years can be helpful in shaping families’ eating habits and improving parental nutrition knowledge. However, the program was less effective in eliminating the respondents’ preference for sweet-tasting foods.

## 1. Introduction

Effective strategies for improving eating habits and dietary intake in toddlers and preschoolers are essential for reducing the risk of chronic non-infectious diseases in later life [1,2]. Unhealthy dietary habits contribute to the high incidence of chronic lifestyle diseases, including overweight and obesity, diabetes, cancer, hypertension and cardiovascular disorders in childhood and adulthood [3,4,5].

Parents and caretakers should instill healthy eating habits in their children, and they should receive support in the process of preparing and eating healthy meals in the family setting [1,6,7,8]. Nurseries and kindergartens play an increasingly important role in family life, and they can contribute to the development of positive dietary habits because most children attend pre-primary educational care institutions [9]. According to Tatlow-Golden et al. (2013), 4-year-olds begin to understand the relationship between health and nutrition [10] and 5- to 6-year-olds are able to make the distinction between healthy and unhealthy foods [2,11,12]; however, these skills do not always lead to healthy food choices, in particular among younger children [13]. Nutritional health is usually a part of standard curricula in early childhood education, and when this topic is addressed regularly and combined with guided practice during meals, it can promote healthy eating habits in children and parents [14].

According to most researchers, parents’ nutritional knowledge significantly influences children’s eating behaviors and food choices [15,16,17]. Most educational programs feature short-term nutrition courses for parents, events promoting healthy eating behaviors, and preschool intervention activities encouraging the consumption of fruit and vegetables as snacks and emphasizing the importance of adequate water intake [18,19,20,21]. However, there is a general scarcity of research focusing on the relationship between nutrition education for children, improvement in parental nutrition knowledge, and the implementation of healthy eating habits in daily practice.

Therefore, the aim of this study was to evaluate the effect of long-term nutrition education for 3- to 6-year-olds on parental nutrition knowledge. The nutrition education program lasted 7 months and comprised six meetings dedicated to different topics. An attempt was made to determine whether long-term multi-stage nutrition education programs that are adapted to the cognitive abilities of preschoolers from different age groups [22] can contribute to an improvement in parental nutrition knowledge and modify the parents’ food preferences and eating habits. Parents’ eating habits that are easiest to modify as well as behaviors that are more firmly established and less susceptible to change were also identified during the study.

## 2. Materials and Methods

### 2.1. Study Design and Participants

The study was conducted as part of the “Colorful Eating is Healthy Eating” nutrition education program that has been implemented in kindergartens in Lublin, a city in south-eastern Poland, since 2016 [23]. The program was introduced in kindergartens belonging to a municipal network of health-promoting schools and kindergartens as well as kindergartens that do not belong to that network but are managed by the local education authorities in Lublin. A total of 11 kindergartens were involved in this stage of the program (the participants were recruited between April and June 2018), and 733 parents consented to participate in the project. The inclusion criteria were: age of 3 to 6 years, absence of metabolic disorders that require an elimination diet or foods for special medical purposes, absence of food allergies and type 1 diabetes. The research tool was an original 54-item questionnaire that was developed for the Polish population based on the Infant and Young Child Feeding (IYCF) assessment and the KomPAN questionnaire for analyzing dietary beliefs and habits [24,25]. In the first part of the questionnaire, parents or caretakers were asked to indicate the child’s age, body weight and height, and to provide information about their own age, gender, education and place of residence. The second part contained questions about the frequency of consumption of specific food groups, and the following options were provided: [] never, [] 1–3 times a month, [] once a week, [] several times per week, [] once daily, [] several times per day. The third part of the questionnaire was designed to examine the parents’ knowledge about healthy nutrition, dietary sources of major macronutrients, food storage and labeling, health benefits of drinking water and health effects of highly processed foods. All parents and caretakers completed the questionnaire voluntarily and consented to their children’s participation in the study. The parents were told that the questionnaire was anonymous, and they were informed about the purpose of the research. The questionnaire was completed independently by the participants without the researchers’ assistance or intervention.

### 2.2. Nutrition Knowledge

The third part of the questionnaire was composed of 15 items analyzing parental nutrition knowledge. The respondents could answer each item with one of three options: true, false or don’t know. One point was awarded for each correct answer, and no points were given for incorrect answers or “don’t know” responses. Points were summed up for each respondent (range 0 to 15 points). Based on the distribution of points, the respondents were divided into three nutrition knowledge categories: low (0–7 points), average (8–11 points) and high (12–15 points).

### 2.3. Data Collection

The study was divided into three stages. In the first stage (September–October 2018), all parents (733 persons) who consented to participate in the project completed a questionnaire containing 54 items. The respondents returned 521 correctly filled questionnaires; 175 respondents did not return the questionnaire or filled it in incorrectly, and 36 respondents withdrew from the project.

In the next stage, 211 children from 4 randomly selected kindergartens participated in the “Colorful Eating Is Healthy Eating” educational program. The program consisted of 6 meetings between November 2018 and May 2019. Each month, preschoolers attended one meeting lasting 30–60 min (Table 1). The meetings relied on expository teaching methods, and they involved puzzles, quizzes, sticker charts, jigsaw puzzles, board games and short cooking workshops with food sampling. All educational materials were approved by kindergarten educators and were adapted to the learning characteristics of children at different developmental levels.

In the third stage of the study (June 2019), the parents (521 persons) of children participating in the educational program and control group children once again completed the 54-item questionnaire. An additional workshop on the healthy food and lifestyle pyramid was held in June 2019 for all children who had not participated in the educational program.

### 2.4. Data Analysis

Categorical variables were presented as sample percentages (%), and continuous variables were expressed by median values and the interquartile range (IQR). The differences between groups were analyzed in the chi-squared test (categorical variables). The Wilcoxon signed-rank test for two dependent samples was used to comprehensively compare test data and retest data and to verify differences in mean food consumption frequency between the test and the retest.

Eating habits and food consumption were presented with the use of qualitative data, and the frequency with which different foods were consumed was expressed on a scale of “never” to “several times a day”. Identical ranks and/or indicators of daily consumption frequency (times/day) were applied to process and interpret the results (Table 2).

The odds ratios (ORs) and 95% confidence intervals (95% CIs) were calculated. The reference categories (OR = 1.00) were the child’s age (5–6 years), parents’ age (26–35 years), education (university education), professional status (permanently employed) and parental nutrition knowledge scores (8–11 points). The ORs were adjusted for parental knowledge about healthy nutrition and the role of nutrients in health maintenance. The significance of ORs was assessed by Wald’s statistics. The results of all tests were regarded as statistically significant at *p* < 0.05. Data were processed in the Statistica program (version 13.1 PL; StatSoft Inc., Tulsa, OK, USA; StatSoft, Krakow, Poland).

## 3. Results

The respondents returned 521 fully and correctly completed questionnaires. In the studied population, 83.7% of the respondents were female (Table 3).

### 3.1. Nutritional Behavior and Dietary Intake

The families’ eating habits and intake of different food products were compared before and after the 6-month educational program for children. The comparison revealed that the educational program induced minor changes in the parents’ dietary preferences.

The consumption of white bread and fine groats decreased (*p* < 0.05) (Table 4 and Table 5). After the program, the respondents from the surveyed group increased their intake of raw and boiled vegetables (*p* < 0.05), but the consumption of pickled and fermented vegetables decreased despite the fact that the health benefits of fermented foods had been discussed during the workshops and that the children consume such products in the kindergarten. A significant (*p* < 0.05) increase was noted in the consumption of fruit and vegetable juice which should be part of a diversified diet, but should not be consumed in quantities exceeding 220 mL/day. A positive outcome of the educational program was that it contributed to a decrease in the consumption of sweetened hot beverages (*p* = 0.014) and an increase in water intake (*p* = 0.001). Before the program, average water intake was 237 ± 254 mL/day in the entire population. In the surveyed group, water intake increased from 311 ± 231 mL/day before the program to 467 ± 313 mL/day after the program. None of the children drank water less than several times a week (the water intake coefficient increased from 0.14 before the program (once a week) to 0.5 after the program (several times a week), which suggests that some children had been drinking water only once a week before the program). The nutrition education program was also successful in reducing the consumption of sweets. Before the program, more than half of the children from the surveyed group (52.3%) ate sweets once a day on average, 45% ate sweets several times a day and only 2.7% ate sweets less than once a week. This percentage was reduced to 34% after the program (*p* < 0.05).

### 3.2. Nutrition Knowledge and Contributing Factors

The parents’ knowledge about healthy dietary patterns and the role of nutrition in maintaining health, as well as changes in parental nutrition knowledge after the educational program were analyzed in the control group and in the surveyed group (Table 6).

The average score was significantly lower in the control group after the educational program than in the surveyed group (*p* < 0.01). After the program, 72% of control group parents demonstrated high levels of nutrition knowledge with an average score of 12.6 ± 1.1 points, whereas 37.4% of the parents from the surveyed group had high levels of nutrition knowledge with an average score of 13.9 ± 0.6 points.

The child’s age was a factor that significantly contributed to the parents’ nutritional knowledge after the program. Five- to 6-year-olds who participated in the program probably conveyed more information to their parents, made better use of the acquired skills, knowledge and recipes, and their parents had higher nutrition knowledge scores (10.6 points on average; range: 8–15 points) than the parents of younger children (9.2 points on average; range: 4–13 points; *p* = 0.0011). The parents’ educational attainment was also a significant contributing factor. Better educated parents had significantly higher nutrition knowledge scores (*p* = 0.021), and 74% of the parents in the surveyed group had high scores (*p* = 0.0008) after the program.

The influence of parental factors on the parents’ knowledge about food products and dietary guidelines for preschool children was analyzed (Table 7). Both before and after the program, parents of younger children (OR 1.76; 95% CI 1.23–1.9, *p* < 0.01), parents older than 35 (OR 1.37; 95% CI 1.2–1.56, *p* < 0.05) and parents with high nutrition knowledge scores (OR 1.78; 95% CI 1.61–1.94, *p* < 0.01) were familiar with dairy intake recommendations for preschoolers.

Before the program, a higher number of parents with primary school education were of the opinion that fast foods can be introduced to their children’s diets similarly to other supplementary foods without increasing the risk of lifestyle diseases (OR 1.56; 95% CI 1.11–1.86, *p* < 0.05). After the program, 71% of parents from the above group correctly indicated that fast foods should not be incorporated into their children’s diets.

Nutrition fact labels provide consumers with important information about the quality of consumed foods. Food labels were very rarely or never read by parents with vocational education (OR 0.54; 95% CI 0.32–0.71, *p* < 0.001), unemployed parents who were homemakers (OR 0.66; 95% CI 0.51–0.76, *p* < 0.01) and parents with low nutrition knowledge scores (OR 0.59; 95% CI 0.49–0.73, *p* < 0.01). After the program, significantly fewer parents with vocational education (OR 0.78; 95% CI 0.49–0.89, *p* < 0.05) and unemployed parents/homemakers (OR 0.77; 95% CI 0.61–0.94 *p* < 0.05) declared that they did not read food labels and that nutritional information was not important for the child’s health.

Children’s education also improved the parents’ knowledge about dietary sources of fiber and the recommended fiber intake, and it contributed to the awareness that breakfast is the most important meal of the day. The program did not enhance the parents’ knowledge about snacking between meals or the role of sweetened beverages in dental caries, overweight and obesity. The parents’ knowledge about the recommended number of fruit and vegetable servings per day was similar in all groups before and after the educational program (*p* > 0.05).

## 4. Discussion

Nutrition education offered as part of the standard preschool curriculum as well as during extramural courses and workshops plays a very important role in a child’s development, and it can contribute to positive changes in families’ eating habits and parents’ nutrition attitudes.

Most studies focus on nutrition education for parents or educational programs targeting both preschoolers and their parents [26,27], but there is a general scarcity of research on nutrition education for 3- to 6-year-olds whose eating habits and food preferences are easily modified through imitation and group learning. Healthy eating habits that have been instilled in the preschool period affect later stages of life and minimize the risk of nutrition-related problems in adulthood, because nutrition affects a child’s physical, social and emotional development as well as behavior [2,13]. The extent to which long-term nutrition education programs for children can improve parental nutrition knowledge and promote positive changes in families’ dietary behaviors should also be evaluated. A limited number of dietary intervention studies have demonstrated that nutrition knowledge and dietary habits can be improved by educational programs targeting only children as well as programs that address both children and their parents [28,29]. According to Kozłowska-Wojciechowska et al. [30], educational programs designed for children and youths are an effective indirect tool for improving parental knowledge about nutrition. Children can influence their parents’ purchasing behaviors [31], and girls do it more often and choose healthier food products [32]. The current study revealed that multi-stage nutrition education can improve general knowledge about healthy dietary behaviors and increase the consumption of health-promoting food ingredients. Long-term educational and promotional campaigns can motivate individuals to change their eating habits, thus instilling social change.

Nutrition fact labels enable consumers to make healthy and informed choices when shopping for food. The present study demonstrated that nutrition education improves label comprehension and encourages consumers to read food labels, thus contributing to an improvement in their eating habits and food preferences [33,34]. The impact of nutrition information on children’s and adults’ health is determined by the extent to which consumers read and understand food labels. [35]. In this study, children who were educated about nutrition fact labels effectively conveyed that knowledge to their parents. Parents who read food labels can use the resulting information to decrease their children’s sugar and calorie intake, thus reducing the risk of dental caries and excessive weight gain [36,37].

For a nutrition education program to be effective, teaching methods and class time should be adapted to the cognitive and behavioral abilities of specific age groups. Most long-term programs consist of classes or meetings that are held at regular intervals (two weeks to one month), which enables the educators to introduce new topics gradually and systematize the imparted knowledge [38,39,40]. According to research for dietary interventions to deliver the anticipated outcomes, nutrition education programs should last 6 months or longer [38,39]. Preschoolers begin to understand concepts such as nutritional value, the role of nutrients and the influence of nutrition on health [2,12,40]. Expository and activating teaching methods that aim to modify the students’ behavior (workshops, group play, role playing) are more effective than strategies that merely covey information (educational talks).

Story telling is one of the most effective teaching methods in preschool education. Stories and characters that relate to children’s daily experiences promote better understanding of the imparted knowledge, and visual aids such as a health food pyramid or a healthy eating plate help children to remember and recall the learned content [41,42]. Educational materials, including sticker charts, cartoons, activity sheets, games and coloring pages, as well as workshops and activities during which children sample new foods, identify differently colored fruits and vegetables, and discover new flavors and tastes play a very important role in nutrition education [20,43]. According to research, preschoolers have a short attention span, and nutrition education classes should last 30 to 45 min [2,14,20,22,41].

Well-designed nutrition education for young children can improve a family’s eating habits. Nutrition education classes for 3- to 6-year-olds that are held in the kindergarten setting do not require parents’ or caretakers’ involvement, and the imparted knowledge can be reinforced through various activities. Young children are highly susceptible to the content of educational curricula, and they are not yet highly influenced by their peers and the environment. The knowledge imparted during educational programs in kindergartens reaches the parents indirectly, and it can improve eating habits in families that do not have healthy dietary behaviors and are unaware that they require assistance in this respect.

In children aged 3 to 6 years, higher water intake could be helpful in reducing the consumption of sweetened hot beverages, fruit juice and carbonated drinks, thus decreasing preschoolers’ daily calorie intake [44,45,46]. According to the American Academy of Pediatrics and the European Society for Pediatric Gastroenterology Hepatology, plain water should be promoted as the principal source of hydration for children [47,48]. Myszkowska-Ryciak [49] reported that nutrition education programs for preschool teachers and other staff members can induce positive changes in the types of beverages served to children in kindergartens and reduce children’s intake of sweetened hot beverages. In the present study, water intake increased significantly, and the consumption of sweetened hot beverages decreased after the 6-month nutrition program. This is a very important observation in view of the results of the ToyBox study which demonstrated that Polish and Belgian preschoolers were characterized by the highest intake of sweet beverages and the lowest water intake in the EU [50].

### Strengths and Limitations

One of the greatest strengths of this study is that the type of activities in the nutrition education program and class time were adapted to the learning capabilities of 3- to 6-year-olds. All educational materials were approved by preschool educators who also assisted the researchers in selecting the most appropriate expository teaching methods. Educational materials were further refined during the program, and they were handed over to the teachers for use in groups that did not participate in the project. Ready-made class scenarios and teaching aids can be helpful in expanding the content of standard preschool curricula relating to healthy nutrition and the promotion of healthy eating habits.

The study also has several weaknesses. The number of respondents who participated in the nutrition education program was relatively small, and the program consisted of only 6 meetings. Kindergarten principals were unable to incorporate more than one nutrition class per month in their schedules, and the number of researchers participating in the project was insufficient to teach more classes.

## 5. Conclusions

Long-term multi-stage nutrition education for children aged 3 to 6 years can be helpful in shaping families’ eating habits and improving parental nutrition knowledge.

Nutrition education programs that rely on expository teaching methods can be effective tools for instilling healthy eating habits in preschoolers and shaping their food preferences provided that class time and the volume of imparted knowledge are adapted to the learning characteristics of children at different developmental levels.

Age is an important factor that significantly influences a child’s ability to transfer the acquired knowledge and skills to the home environment. After the program, the parents of older children had higher nutrition knowledge scores than the parents of younger children.

The nutrition education program contributed to more frequent choices of health-promoting food products and increased water intake in families. However, the program was less effective in eliminating the respondents’ preference for sweet-tasting foods.

Educational campaigns on how to use nutrition fact labels should also target children who have not yet learned to read to promote healthy food buying habits and raise awareness that food products differ in quality and health safety.

## Figures and Tables

**Table 1 ijerph-19-01981-t001:** Topics and content of nutrition education meetings.

	Topic	Content	Teaching Method
1	Healthy food and lifestyle pyramid (duration: 60 min)	− Role of the healthy food pyramid; − What is the meaning of each level in the pyramid? − Importance of physical exercise and sleep	Building a healthy food pyramid with the use of food packaging; creating a food pyramid collage (to be taken home); making a food pyramid out of felt
2	Fruit and vegetables—colorful and healthy (duration: 60 min)	− Health-promoting components in fruit and vegetables; − Why do fruits taste sweet? − Eating 5 servings of fruit and vegetables each day	Puzzles and quizzes (why fruits are colorful); vegetable salad
3	How to build strong teeth and bones (duration: 45 min)	− Calcium-rich foods; − Are all types of milk equally healthy? − Sweetened dairy foods and replacements	Educational poster on sources of calcium; Strong Bones jigsaw puzzle
4	Can we survive without eating meat? (duration: 30 min)	− Is a vegetarian diet better than a traditional diet: − Health benefits of eating less red meat; − What is the role of iron?	Puzzles; work sheets—vitamins and minerals in different types of meat
5	Sweet preschoolers (duration: 45 min)	− Preference for sweet-tasting foods; − Healthy foods in the diet; − Are all sweets unhealthy?	Recipes for home-made ice-cream and whole-grain cookies; analyzing the content of sugar in food products
6	How about water? (duration: 45 min)	− Role of water in the body; − How much sugar is there in flavored water and juice? − How to make water taste better?	Recipes for home-made lemonade; crossword puzzle about water

**Table 2 ijerph-19-01981-t002:** Ranks and/or indicators of daily consumption frequency (times/day) which applied to process and interpret the results of eating habits and food consumption.

Consumption Frequency Categories	Ranks Assigned to Consumption Frequency Categories	Daily Consumption Frequency (Times/Day)
never	1	0
1–3 times per month	2	0.06
once a week	3	0.14
several times a week	4	0.5
once a day	5	1
several times a day	6	2

**Table 3 ijerph-19-01981-t003:** Characteristics of the studied population.

	Total Sample, *n* = 521	Surveyed Group, *n* = 211	Control Group, *n* = 310	*p*
Child’s age, *n* (%)				
3–4 years	236 (45.3)	98 (46.4)	138 (44.5)	ns
5–6 years	285 (54.7)	113 (53.6)	172 (55.5)	
Gender, *n* (%)				
Female	436 (83,7)	177 (83.9)	259 (83.5)	
Male	85 (16.3)	34 (16.1)	51 (16.5)	0.03
Parents’ age, *n* (%)				
<25 years	44 (8.4)	23 (10.9)	21 (6.8)	
26–35 years	351 (67.4)	136 (64.5)	215 (69.4)	0.034
>35 years	126 (24.2)	52 (24.6)	74 (23.8)	
Place of residence, *n* (%)				
City > 100,000	313 (60.1)	127 (60.2)	186 (60.0)	
City < 100,000	132 (25.3)	48 (22.7)	84 (27.1)	0.002
Rural area	76 (14.6)	36 (17.1)	40 (12.9)	
Education				
Primary school	21 (4.0)	7 (3.3)	14 (4.5)	
Secondary school	139 (26.7)	41 (19.4)	98 (31.6)	0.017
Vocational school	42 (8.1)	13 (6.2)	29 (9.4)	
University	319 (61.2)	150 (71.1)	169 (54.5)	
Professional status				
Unemployed, homemaker	60 (11.5)	23 (10.9)	37 (11.9)	
Works occasionally	174 (33.4)	49 (23.2)	125 (40.3)	0.036
Permanently employed	256 (49.1)	131 (62.1)	125 (40.3)	
University student	31 (6.0)	8 (3.8)	23 (7.5)	

The results are statistically significant at *p* < 0.05 (chi-squared test).

**Table 4 ijerph-19-01981-t004:** Eating habits in the family in the studied population and in the surveyed group before and after the 6-month nutrition education program.

	Total Sample	Surveyed Group before the Program	Surveyed Group after the Program	*p*
Eating habits				
Number of meals per day	4 (2–7)	4 (2–6)	4 (2–6)	
Snacking between meals	0.8 (0–2)	0.7 (0–2)	0.7 (0–2)	
Adding sugar to beverages	0.45 (0–0.5)	0.44 (0–0.5)	0.42 (0–1)	0.008
Adding salt to meals	0.37 (0–0.5)	0.27 (0–0.5)	0.25 (0–0.5)	

**Table 5 ijerph-19-01981-t005:** Comparision of food consumption frequency (times/day) of various food products in the studied population and in the surveyed group before and after the 6-month nutrition education program.

	Total Sample before Program (Test)	Total Sample after the Program (Retest)	*p*	Surveyed Group before the Program (Test)	Surveyed Group after the Program (Retest)	*p*
Food consumption frequency						
Whole-grain bread	0.55	0.64	0.021	0.61	0.64 (0–2)	0.08
Wheat bread	0.83	0.67	0.012	0.74	0.52 (0–2)	0.002
Coarse groats (buckwheat, cereal flakes)	0.47	0.57	0.017	0.56	0.56 (0–2)	0.324
Fine groats, rice	0.43	0.40	0.174	0.48	0.41 (0–1)	0.031
Milk	0.71	0.66	0.071	0.68	0.67	0.214
Fermented milk beverages	0.46	0.51	0.326	0.6	0.58	0.314
Cottage cheese, tvorog	0.48	0.51	0.216	0.51	0.53	0.209
Hard and soft ripened cheese	0.62	0.63	0.346	0.55	0.54	0.190
Red meat (e.g., beef)	0.41	0.44	0.522	0.48	0.46	0.227
White meat (e.g., chicken and turkey)	0.65	0.63	0.372	0.59	0.59	0.617
Fish	0.26	0.31	0.276	0.34	0.38	0.07
Eggs	0.61	0.58	0.346	0.55	0.55	0.510
Legumes (e.g., peas, beans, lentils)	0.22	0.27	0.072	0.27	0.34	0.031
Vegetables	0.71	0.79	0.041	0.71	0.83	0.027
Fermented vegetables	0.39	0.38	0.517	0.42	0.39	0.167
Fruits	0.71	0.75	0.278	0.76	0.82	0.017
Fruit and vegetable juice	0.59	0.63	0.314	0.31	0.38	0.024
Butter	0.47	0.51	0.241	0.69	0.69	0.511
Margarine	0.57	0.67	0.071	0.57	0.69	0.017
Sweetened hot beverages	0.86	0.83	0.317	0.79	0.71	0.011
Fast food	0.31	0.28	0.176	0.26	0.28	0.314
Sweetened beverages	1.00	0.84	0.021	1.00	0.78	0.001
Water	0.68	0.77	0.001	0.81	0.91	0.001

*p* < 0.05 significance level of Wilcoxon’s test (for two dependent samples) for differences in means of food consumption frequency (times/day) between the test and the retest.

**Table 6 ijerph-19-01981-t006:** Nutrition knowledge scores (in points) and factors contributing to differences in parental nutrition scores in the overall population, in the control group and in the surveyed group before and after the 6-month educational program.

	Total Sample	Before the Program (Baseline)		*p*	After the Program (Follow-Up)		*p*
		Control Group	Surveyed Group		Control Group	Surveyed Group	
	Nutrition Knowledge Score (Points)						
Average score	8.4 (5.9–9.2)	8.1 (6.1–8.6)	8.6 (7.4–10.5)	ns	8.3 (6.7–10.1)	9.5 (7.9–12.3)	0.008
	Nutrition Knowledge Score, *n* (%)						
low	136	79	63		84	79	
average	136	114	74		116	74	
high	249	117	74		110	58	
	Child’s Age						
3–4 years	8.2	8.2	8.1	ns	8.2	9.2	0.005
5–6 years	8.5	8.0	8.8	0.043	8.4	10.6	0.0001
	Parents’ Education						
Primary school	6.7	6.3	6.8	ns	6.8	6.9	ns
Secondary school	8.5	8.7	8.2	0.041	9.2	8.7	0.04
Vocational school	7.8	7.7	7.9	ns	7.9	8.1	ns
University	9.3	9.1	9.6	ns	9.4	12.4	0.0001

Low knowledge score (0–7 points), average knowledge score (8–11 points), high knowledge score (12–15 points), ns—not significant; Changes were calculated as the difference between follow-up vs. baseline values within one group (control or surveyed); *p*-value—significance level for the observed difference (Mann–Whitney test) or change (Wilcoxon test).

**Table 7 ijerph-19-01981-t007:** Odds ratios (95% confidence interval) in an analysis of the relationships between parental knowledge about healthy eating and the role of nutrition in health maintenance vs. child’s age, parents’ age, education, professional status and nutrition knowledge score.

	Child’s Age (Ref. 5–6 Years) 3–4 Years	Parents’ Age (Ref. 26–35 Years)		Parents’ Education (Ref. University Education)			Professional Status (Ref. Permanently Employed)			Nutrition Knowledge (Ref. Average)	
		>25 Years	<35 Years	Primary	Secondary	Vocational	Unemployed/ Homemaker	Works Occasionally	University Student	Low	High
Before the Program											
Dairy products	1.76 ** (1.23–1.9)	1.02 (0.9–1.09)	1.37 * (1.2–1.56)	0.62 ** (0.51–0.73)	0.92 (0.8–0.94)	0.92 (0.74–0.98)	0.91 (0.84–1.03)	0.74 * (0.7–0.91)	1.06 (0.92–1.14)	1.03 (0.94–1.22)	1.78 ** (1.61–1.94)
Fiber	0.89 (0.77–0.94)	1.04 (0.89–1.09)	0.94 (0.78–1.08)	0.54 ** (0.31–0.7)	1.04 (0.91–1.11)	0.71 * (0.63–0.82)	0.88 (0.74–0.94)	0.91 (0.77–1.01)	1.02 (0.91–1.09)	0.88 (0.71–1.07)	1.43 * (1.22–1.7)
Vegetables	0.81 * (0.74–0.93)	0.94 (0.91–0.98)	1.12 (1.03–1.19)	1.05 (1.01–1.09)	0.93 (0.88–0.97)	0.98 (0.91–1.03)	1.03 (1.01–1.06)	1.11 (1.05–1.17)	0.94 (0.83–1.08)	0.91 (0.83–0.96)	1.09 (1.02–1.14)
Fruits	1.09 (1.02–1.17)	0.97 (0.88–1.08)	1.02 (0.96–1.09)	1.12 (1.03–1.14)	0.99 (0.92–1.04)	1.02 (0.91–1.12)	1.08 (1.01–1.12)	1.02 (0.91–1.06)	1.12 (1.07–1.15)	1.02 (0.94–1.05)	1.07 (0.98–1.12)
Fast food	0.77 * (0.62–0.91)	0.91 (0.84–0.98)	0.74 * (0.56–0.89)	1.56 * (1.11–1.86)	1.01 (1.0–1.04)	1.37 * (1.24–1.49)	0.61 ** (0.56–0.77)	1.98 ** (1.57–2.21)	1.02 (0.96–1.09)	1.01 (0.96–1.09)	0.74 * (0.58–0.89)
Breakfast	2.11 ** (1.74–2.23)	0.95 (0.9–0.99)	1.02 (0.91–1.06)	1.09 (1.04–1.19)	1.02 (0.94–1.14)	1.04 (0.96–1.09)	1.48 * (1.27–1.63)	0.71 * (0.63–0.89)	0.95 (0.89–1.09)	1.05 (1.01–1.11)	1.37 * (1.22–1.45)
Snacking between meals	0.73 * (0.61–0.93)	0.89 (0.85–0.92)	1.13 (1.03–1.17)	1.02 (0.93–1.07)	0.97 (0.91–1.02)	1.07 (1.01–1.11)	1.06 (1.02–1.09)	1.04 (0.96–1.08)	1.06 (0.93–1.11)	0.71 * (0.64–0.89)	1.08 (1.01–1.14)
Sweetened beverages	1.56 * (1.21–1.77)	1.09 (1.02–1.17)	1.02 (0.9–1.11)	1.08 (1.04–1.15)	0.92 (0.84–0.99)	1.27 * (1.15–1.39)	0.94 (0.88–0.08)	1.05 (0.97–1.09)	1.09 (1.02–1.16)	0.99 (0.95–1.03)	0.86 * (0.79–0.93)
Reading food labels	1.06 (0.94–1.17)	1.05 (0.99–1.09)	1.05 (1.0–1.12)	1.11 (1.09–1.17)	0.95 (0.89–1.02)	0.54 ** (0.32–0.71)	0.66 ** (0.51–0.76)	0.7 * (0.67–0.81)	1.07 (0.99–1.12)	0.59 ** (0.49–0.73)	1.79 ** (1.55–1.89)
After the Program											
Dairy products	1.52 * (1.2–1.76)	1.02 (0.9–1.09)	1.56 * (1.31–1.89)	0.62 ** (0.51–0.73)	0.92 (0.8–0.94)	0.92 (0.74–0.98)	0.91 (0.84–1.03)	0.74 * (0.7–0.91)	1.06 (0.92–1.14)	1.03 (0.94–1.22)	1.84 ** (1.65–1.99)
Fiber	0.89 (0.77–0.94)	1.04 (0.89–1.09)	0.94 (0.78–1.08)	0.71 * (0.56–0.84)	1.04 (0.91–1.11)	0.71 * (0.63–0.82)	0.88 (0.74–0.94)	0.91 (0.77–1.01)	1.02 (0.91–1.09)	0.88 (0.71–1.07)	1.43 * (1.22–1.7)
Vegetables	0.81 * (0.74–0.93)	0.94 (0.91–0.98)	1.12 (1.03–1.19)	1.05 (1.01–1.09)	0.93 (0.88–0.97)	0.98 (0.91–1.03)	1.03 (1.01–1.06)	1.11 (1.05–1.17)	0.94 (0.83–1.08)	0.91 (0.83–0.96)	1.09 (1.02–1.14)
Fruits	1.09 (1.02–1.17)	0.97 (0.88–1.08)	1.02 (0.96–1.09)	1.12 (1.03–1.14)	0.99 (0.92–1.04)	1.02 (0.91–1.12)	1.08 (1.01–1.12)	1.02 (0.91–1.06)	1.12 (1.07–1.15)	1.02 (0.94–1.05)	1.07 (0.98–1.12)
Fast food	0.77 * (0.62–0.91)	0.91 (0.84–0.98)	0.56 ** (0.43–0.71)	0.98 (0.91–1.06)	1.01 (1.0–1.04)	1.37 * (1.24–1.49)	0.61 ** (0.56–0.77)	1.98 ** (1.57–2.21)	1.02 (0.96–1.09)	1.01 (0.96–1.09)	0.74 * (0.58–0.89)
Breakfast	1.89 ** (1.51–2.11)	0.95 (0.9–0.99)	1.02 (0.91–1.06)	1.09 (1.04–1.19)	1.02 (0.94–1.14)	1.04 (0.96–1.09)	1.48 * (1.27–1.63)	0.86 (0.61–0.94)	0.95 (0.89–1.09)	1.05 (1.01–1.11)	1.37 * (1.22–1.45)
Snacking between meals	0.73 * (0.61–0.93)	0.89 (0.85–0.92)	1.13 (1.03–1.17)	1.02 (0,93–1.07)	0.97 (0.91–1.02)	1.07 (1.01 –1.11)	1.06 (1.02–1.09)	1.04 (0.96–1.08)	1.06 (0.93–1.11)	0.71 * (0.64–0.89)	1.08 (1.01–1.14)
Sweetened beverages	1.56 * (1.21–1.77)	1.09 (1.02–1.17)	1.02 (0.9–1.11)	1.08 (1.04–1.15)	0.92 (0.84–0.99)	1.27 * (1.15–1.39)	0.94 (0.88–0.08)	1.05 (0.97–1.09)	1.09 (1.02–1.16)	0.99 (0.95–1.03)	0.86 * (0.79–0.93)
Reading food labels	1.06 (0.94–1.17)	1.05 (0.99–1.09)	1.05 (1.0–1.12)	1.11 (1.09–1.17)	0.95 (0.89–1.02)	0.78 * (0.49–0.89)	0.77 * (0.61–0.94)	0.7 * (0.67–0.81)	1.07 (0.99–1.12)	0.59 ** (0.49–0.73)	1.79 ** (1.55–1.89)

The results are statistically significant at: * *p* < 0.05; ** *p* < 0.01 (Wald’s statistics).

## Data Availability

Due to ethical restrictions and participant confidentiality, data cannot be made publicly available. All data were collected in an anonymous way. The data that support the findings of this study are available from the authors upon reasonable request.

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
