# Peer review of "The Effect of the “Colorful Eating Is Healthy Eating” Long-Term Nutrition Education Program for 3- to 6-Year-Olds on Eating Habits in the Family and Parental Nutrition Knowledge"

_ijerph, 2022, doi:10.3390/ijerph19041981_

Round 1
Reviewer 1 Report
- Revisit the study keywords. (choose keywords that will give high search volume).
- Aim is not clear. try to rephrase.
- In methodology section, explain what did you use to interpret the studies in language other than English (since your study presentation is in English).
- Discussion section. Make sure you discuss all your findings. Provide context and explain why people (reader) should care. and do not introduce new results. do not include recommendation in discussion section (check paragraph line 251, and 262).
- Strength & limitation. rephrase study weakness (check paragraph line 294), describe why the study limitation/ weakness could not be overcome.
Author Response
Thank you very much for your time and insightful and detailed review that helped us improve our manuscript. All changes are highlighted blue font.
|
|
pages |
Revisit the study keywords. (choose keywords that will give high search volume). |
The keywords were revised to target those with a high search volume. |
30-31 |
Aim is not clear. try to rephrase. |
The aim of the study was described in greater detail. |
59-67 |
In methodology section, explain what did you use to interpret the studies in language other than English (since your study presentation is in English). |
I am not quite sure I understand this comment. The studies referenced in the methodology section are in English and Polish, and I am a native speaker of Polish. |
|
Discussion section. Make sure you discuss all your findings. Provide context and explain why people (reader) should care. and do not introduce new results. do not include recommendation in discussion section (check paragraph line 251, and 262). |
Thank you for bringing this to my attention. The recommendations and references to my previous studies were removed from the Discussion section.
|
301-309 318-319 |
Strength & limitation. rephrase study weakness (check paragraph line 294), describe why the study limitation/ weakness could not be overcome. |
The limitations of the study were rephrased. |
333-337 |

Reviewer 2 Report
The study at hand deals with the important topic of how to make nutrition guidelines effective by educating small children. The author found some effects of the education program on both the eating habits and the knowledge on nutrition, but the programme did not achieve all desired changes. The novelty of the presented approach lies in the fact that 3-6 year old children supposedly educate their parents on healthy nutrition.
The manuscript is clearly structured and written in a professional manner, meeting scientific standards. The English is fine with just a few minor flaws (some preposition errors (e.g. Lines 15 and 16), comma mistakes in line 104).
A major flaw of the study is that the researcher applied the wrong test. She used the Mann-Whitney-U test although the samples were paired (before-after). Instead, Wilcoxon’s test for paired samples should have been applied. Typically, paired tests provide higher significances (lower p values). The application of the wrong statistical test jeopardizes both the Results and the Discussion.
I was disappointed to see that non-scientific views on the impact of nutrition on health (“alkalizing foods”) were part of the education programme and it made me concerned that esoteric, not scientifically proven assumptions on the health impact of food were spread to children, increasing disinformation on healthy nutrition.
Moreover, some of the methods, particularly referring to how the results were calculated and evaluated need to be described in more detail. The precise p values of the statistical tests applied are missing. These details have also to be changed before the manuscript can be recommended for publication.
The Discussion is thorough and proves the comprehensive knowledge of the researcher in this area. However, the Discussion may have to be adapted to the Results after the correct test will have been applied.
In detail:
Title
A title comprising four lines seems to long – please shorten that and avoid unnecessary repetitions (education/educational), e.g. “The influence of the long-term nutrition educational program “Colorful Eating Is Healthy Eating” for children aged 3 to 6 years on changes in families’ dietary habits and parental nutrition knowledge”
Abstract
In the Methods section, the readers should be informed about timespans between the intervention and the third stage.
Introduction
Lines35-37. Add here a reference substantiating this claim.
Lines 48-51. Here, a reference on the effect of curricula on healthy eating habits is missing as well.
Line 62: Please clarify what “long-term” means in the context of this study.
Materials and methods
Lines 69-76: Please indicate a reference to a website or publication describing the “Colorful Eating is Healthy Eating” programme.
Lines 78-81: Again, please add references on the applied two questionnaires. It is unclear who developed them and whether they have been validated before. This is of particular importance as the validity of the entire study depends on the quality of these questionnaires.
Line 95: Which “questionnaire” is meant here?
Line 99: This is not a “tertile” distribution, as a tertile typically depends on the frequency of the chosen answers (depend variable, not the independent one)
Line 106: I think it should say “filled it in incorrectly”.
Lines 113-114 and line 288: I do not think that “materials” can be “consulted”.
Table 1: Under Topic 2, the question should be phrased “Why do fruits taste sweet?“. Under Topic 4, replace the semicolon after “foods” with a quotation mark.
Lines 126-127: Neither the Chi-square test nor the (incorrectly applied) Mann-Whitney-U test require normal distribution (as the MWU-test is a non-parametric rank sum test), so I suggest to delete the sentence starting with “Before statistical analysis…”. The suggested (correct) test of Wilcoxon is non-parametric as well.
Lines 130-131. It is unclear how the OR=1 groups were defined. Was it the youngest age group etc.? The information given is incomplete. Furthermore, the procedure of standardization requires further explanation.
Results
Table 2: What do the asterisks in the table mean? If they indicate significance (of the Chi-square test, I assume), the name of the test should be mentioned in the legend and the precise p values should be provided instead of asterisks.
Line 147: Replace “…and the children…” with “…and that the children…”.
Line 156: Unclear how this “coefficient” was calculated”.
Lines 159-161: Which share of the children ate sweets once a day and how did this share change after the intervention?
Table 3: It is unclear what the numbers in the table mean and how they were calculated. The explanation of the meaning of one or two asterisk is missing and, again, the provision of precise p values is mandatory. The legend of the table should indicate which test was applied to obtain the p values and which groups were included in the test (e.g. before vs. after). It is unclear whether the “Total sample” was included in statistical sampling (which should not be the case). The meaning of the values in the parentheses needs to be explained in the legend as well.
Table 4: Provide precise p values and indicate in the legend which test was applied and which of the indicated groups were included in the test. Again, the term “tertile” is used wrongly here – it would indicate that equal shares of respondents belong to each of the tertiles (namely 33.3%), which is seemingly not the case.
Lines 178-186: Were these indicated differences (young vs. old children, higher vs. lower educated parents) significant? If so, at which p levels? Which test was applied to justify the claim “more” and “higher”?
Table 5: What does “Alkalizing foods” mean? Are there any scientific studies proving that these foods have any health impact? I am disappointed to see that such non-scientific views on the impact of nutrition on health is part of an education programme on “healthy” food and it makes me concerned that esoteric, not scientifically proven assumptions on the health impact of food can be spread from professionals in such programs, increasing disinformation on healthy nutrition. For the sake of the trustworthiness of this study, I suggest deleting all information on “alkalizing” foods from the manuscript.
Discussion
Lines 221-227: There is no need to repeat the information already provided in the Introduction.
Author Response
Thank you very much for your time and insightful and detailed review that helped us improve our manuscript. All changes are highlighted red font.
Thank you for a thorough review of the manuscript. The Reviewer’s inputs are highly valuable, and they have enabled me to improve the quality of the article. I agree that the Wilcoxon signed-rank test for two dependent samples should have been used to verify differences in mean food consumption frequency between the test and the retest. Both the Wilcoxon test and the Mann-Whitney test were used in my previous studies. The Reviewer’s remarks were taken into consideration in the revision process, and all changes are marked in red.
|
|
pages |
A title comprising four lines seems to long – please shorten that and avoid unnecessary repetitions (education/educational), e.g. “The influence of the long-term nutrition educational program “Colorful Eating Is Healthy Eating” for children aged 3 to 6 years on changes in families’ dietary habits and parental nutrition knowledge” |
The title was shortened. |
2-3 |
In the Methods section, the readers should be informed about timespans between the intervention and the third stage. |
The third stage of the study was conducted in June 2019, i.e. one month after the nutrition education program had been completed in kindergartens. This date was selected because some kindergartens closed for the summer break on 1 July 2019. The relevant information was provided in the Methods and Abstract sections.
|
17-18 119 |
Lines35-37. Add here a reference substantiating this claim. |
This is a general claim based on a review of the literature. The appropriate reference was provided, as suggested by the Reviewer.
|
36 |
Lines 48-51. Here, a reference on the effect of curricula on healthy eating habits is missing as well. |
50 |
|
Line 62: Please clarify what “long-term” means in the context of this study. |
The program lasted 7 months and comprised 6 meetings dedicated to various topics. The relevant information was provided in the Methods section of the revised manuscript.
|
60-61 |
Please indicate a reference to a website or publication describing the “Colorful Eating is Healthy Eating” programme. |
The program does not have a website. The publication describing the results was referenced in the revised manuscript. |
23 |
Again, please add references on the applied two questionnaires. It is unclear who developed them and whether they have been validated before. This is of particular importance as the validity of the entire study depends on the quality of these questionnaires. |
Thank you for this valuable comment. The questionnaires have been validated; they have been used in other published studies (referenced in the manuscript), and they are recommended by the Committee on Human Nutrition Science of the Polish Academy of Sciences.
|
83 |
Line 95: Which “questionnaire” is meant here? |
I am referring to the same questionnaire (third part) that was discussed in the previous paragraph.
|
|
Line 99: This is not a “tertile” distribution, as a tertile typically depends on the frequency of the chosen answers (depend variable, not the independent one) |
Thank you for spotting this error. The word “tertile” was replaced with “nutrition knowledge categories”.
|
101-103 |
Line 106: I think it should say “filled it in incorrectly”. |
The relevant correction was made, thank you. |
108 |
Lines 113-114 and line 288: I do not think that “materials” can be “consulted”. |
We consulted with kindergarten educators to ensure that the developed materials were adapted to the learning characteristics of children at different developmental levels and that they were consistent with the educational standards for preschoolers. The word “consulted” was replaced with “approved by”.
|
116 |
Table 1: Under Topic 2, the question should be phrased “Why do fruits taste sweet?“. Under Topic 4, replace the semicolon after “foods” with a quotation mark. |
The question under Topic 2was rephrased accordingly. There is no word “foods” under Topic 4.
|
|
Lines 126-127: Neither the Chi-square test nor the (incorrectly applied) Mann-Whitney-U test require normal distribution (as the MWU-test is a non-parametric rank sum test), so I suggest to delete the sentence starting with “Before statistical analysis…”. The suggested (correct) test of Wilcoxon is non-parametric as well. |
Thank you for this observation. The sentence was deleted. The Wilcoxon test was additionally performed, and the results were presented in Table 3 which was thoroughly revised.
|
128-137 |
Lines 130-131. It is unclear how the OR=1 groups were defined. Was it the youngest age group etc.? The information given is incomplete. Furthermore, the procedure of standardization requires further explanation. |
OR=1 groups were defined based on the following parameters: Children’s age: 5-6 years Parents with university education Permanently employed parents Parents with average nutrition knowledge scores
|
138-141 |
Table 2: What do the asterisks in the table mean? If they indicate significance (of the Chi-square test, I assume), the name of the test should be mentioned in the legend and the precise p values should be provided instead of asterisks. |
The precise p values were provided to indicate significance in the chi-squared test (p<0.05). Table 2 was modified. |
150 |
Line 147: Replace “…and the children…” with “…and that the children…”. |
The sentence was modified as suggested. |
159 |
Line 156: Unclear how this “coefficient” was calculated”. |
The coefficient was calculated according to the method proposed by the questionnaire’s authors. The method was described in subsection 2.4 (Data analysis).
|
|
Lines 159-161: Which share of the children ate sweets once a day and how did this share change after the intervention? |
The sentence describes the results in the surveyed group. The relevant information was provided in the revised manuscript.
|
168-174 |
It is unclear what the numbers in the table mean and how they were calculated. The explanation of the meaning of one or two asterisk is missing and, again, the provision of precise p values is mandatory. The legend of the table should indicate which test was applied to obtain the p values and which groups were included in the test (e.g. before vs. after). It is unclear whether the “Total sample” was included in statistical sampling (which should not be the case). The meaning of the values in the parentheses needs to be explained in the legend as well. |
Table 3 and 4 were modified, and precise p values were given. The values in the table were calculated based on the questionnaire interpretation key, and they are described in the methodology developed by the Committee on Human Nutrition Science of the Polish Academy of Sciences. The relevant explanation was provided in subsection 2.4 (Data analysis). The results of the Wilcoxon test are presented in the modified Table 4. The total sample was not included in the statistical analysis, and the results were presented in a comparative approach.
|
|
Table 4: Provide precise p values and indicate in the legend which test was applied and which of the indicated groups were included in the test. Again, the term “tertile” is used wrongly here – it would indicate that equal shares of respondents belong to each of the tertiles (namely 33.3%), which is seemingly not the case. |
Thank you for this observation. The term “tertile” was deleted. Precise p values were given, and additional information was provided under the table. Table 4 was modified.
|
217-218 |
Lines 178-186: Were these indicated differences (young vs. old children, higher vs. lower educated parents) significant? If so, at which p levels? Which test was applied to justify the claim “more” and “higher”? |
Precise p values were provided. |
210, 212, 213 |
Table 5: What does “Alkalizing foods” mean? Are there any scientific studies proving that these foods have any health impact? I am disappointed to see that such non-scientific views on the impact of nutrition on health is part of an education programme on “healthy” food and it makes me concerned that esoteric, not scientifically proven assumptions on the health impact of food can be spread from professionals in such programs, increasing disinformation on healthy nutrition. For the sake of the trustworthiness of this study, I suggest deleting all information on “alkalizing” foods from the manuscript. |
I agree that the alleged health impact of “alkalizing foods” has not been proven in scientific studies. However, some vegetables, in particular root vegetables, have been found to influence the acid-base balance. As suggested, all information on “alkalizing foods” was deleted.
|
|
Lines 221-227: There is no need to repeat the information already provided in the Introduction. |
The repeated sentence on the links between nutrition and diet-dependent diseases was deleted.
|
252-256 |
